# OpenReview forum: "FLASK: Fine-grained Language Model Evaluation based on Alignment Skill Sets"
_ICLR.cc/2024/Conference — ICLR 2024 spotlight_

### Official Review · Reviewer_aEcx · 2023-10-30

**Soundness:** 3 good
**Presentation:** 3 good
**Contribution:** 4 excellent
**Rating:** 8
**Confidence:** 2

**Summary:**

This work proposes an evaluation framework FLASK that includes 12 fine-grained metrics over 4 ability dimensions (logical thinking, background knowledge, problem handling and user alignment) to evaluate the skill set of current large language models (LLMs). Its evaluation results on dataset constructed from MMLU, BBH, FinQA and Haiku Generation dataset provides a more detailed picture of the abilities of LLM that benefits the model choices. For example, Vicuna and Alpaca fall behind GPT-3.5 on logical thinking and background knowledge abilities, and not all abilities receive consistent benefit from scaling. It also splits a hard subset FLASK-hard and finds that proprietary models like GPT4 cannot handle it well.

**Strengths:**

- A new framework for evaluating the skillset of LLMs focused on dimensions like logical thinking and background knowledge that would benefits various reasoning tasks and user alignment that benefits the alignment research by filling the blank of detailed evaluations.
- The difficulty level annotation is informative in differentiating the instances into different levels of ability requirement. And the findings on FLASK-hard reveals the even the strongest proprietary models so far like GPT-4 fall short on solving expert-level problems and highlights the future research directions in enhancing specific models.
- The experimental results are rich and include patterns and biases observed that can be overlooked in the overall metrics.

**Weaknesses:**

- The annotator recruited are people with at least senior undergraduate students, therefore I am not sure whether the difficulty annotations for simple lifestyle knowledge (easy to answer without formal education) and advanced lifestyle knowledge (no formal education but explaining a well-known concept) are fair. Though it may not be a perfect solution for this, I would like to hear the thoughts from the authors about designing such pipeline in an affordable manner.
- The Expert-level knowledge set (FLASK-hard) is not a large set.

**Questions:**

See weakness section.

---

> ### Author Response · Authors · 2023-11-15
> **Rebuttal by authors**
>
> Thank you for taking the time to review our paper. Our responses are shown below.
>
> ---
> **W1. Difficulty annotation by senior undergraduate students**
>
> We acknowledge that there might be issues regarding the distribution of the human labelers affecting the difficulty annotation process. To mitigate this issue, we have provided clear descriptions and examples corresponding to the specific difficulty to clarify the boundaries between different difficulties for both model-based and human-based metadata annotation. The full instructions are specified in Figures 49, 50, and 51 in the Appendix.
>
> ---
> **W2. The small number of instances for FLASK-Hard**
>
> We acknowledge that the number of instances of FLASK-Hard might not be sufficient. This is because we controlled the FLASK whole set to be around 1800 instances to enable academic budget evaluation using model-based evaluation (GPT-4) or human-based evaluation. However, since the metadata annotation process could be fully automated by utilizing LLMs as described in Section 3.2., the FLASK evaluation set could be dynamically expanded without extensive labeling processes, which eventually leads to the increased number of instances consisting of FLASK-Hard after filtering.

---

> > ### Comment · Reviewer_aEcx · 2023-11-22
> >
> > Thanks the authors for the reply.

---

### Official Review · Reviewer_Qvzt · 2023-11-01

**Soundness:** 2 fair
**Presentation:** 1 poor
**Contribution:** 3 good
**Rating:** 8
**Confidence:** 2

**Summary:**

This paper describes a new benchmark for evaluating large language models (LLMs) in a fine-grained fashion, namely FLASK (Fine-grained Language Model Evaluation Based on Alignment Skill Sets). FLASK contains 1,700 example instances compiled from 120 NLP datasets such ash MMLU and GSM8K. It creates a set of four primarily abilities (logical thinking, background knowledge, problem handling, and user alignment), which are further divided into 12 fine-grained skills. Each of the 1,700 instances in FLASK is labeled with regard to the top-3 required skills among the twelve. The authors demonstrate that using fine-grained evaluation of LLM outputs enhance the agreement between GPT-4's automatic rating results and human-rater results. The authors applied FLASK on closed-source LLMs such as GPT-3.5 and Bard, as well as open-source LLMs including Alpaca-13B and Vicuna-13B. The results indicate that open-source LLMs currently lag behind closed-source ones in all the skill categories, but especially in logical thinking and background knowledge.

**Strengths:**

S1. Defining a novel, skill-based, fine-grained method for evaluating the quality of LLMs that is applicable to problems in many domains. The skill breakdown among the four primary abilities and 12 finer-grained abilities capture the most important qualities that users desire in an LLM.
S2. Demonstrating through empirical results that evaluation results show increased human-machine agreement under FLASK's fine-grained approach compared to traditional approaches such as those based on metrics like ROGUE and skill-agnostic evaluation. This advances the state of art for LLM evaluation.

**Weaknesses:**

W1. There seems to be some arbitrariness in creating the list of 12 fine-grained skills. Some of the skills seem to be not clearly motivated or crisply applicable to all problem instances. For instance, the fine-grained skill "readability" applies to *all* problems, because readability of generated text is desirable to the human users regardless of the domain or difficulty of the problem. So this begs the question for what sort of problems would "readability" be singled out as an "essential skill". As another example, the fine-grained skill "metacognition" is related to whether the model is aware of the limits of its own knowledge. Therefore, whether a problem instance requires metacognition seems to depend on whether the problem is beyond the ability of the model, which is of course dependent on the model itself. So it's unclear to me how metacognition could be labeled as the essential skill of a problem instance in a model-independent way. In summary I think there is room for improvement in the details of the four abilities and 12 fine-grained skills, but this doesn't take away from the primary strength of this paper (S1 and S2).
W2. Likewise, there seems to be some arbitrariness in the ten problem domains, in two aspects. First, there are some obvious missing domains such as laws, finance, and travel. It is unclear which of the ten domains those would fall into. Second, it's unclear how this system handles problem instances that involve multiple domains.
W3. The conclusion in the comparison of open- and closed-source LLMs is somewhat dubious because of a lack of public knowledge of the closed-source LLMs' parameter count. The open-source LLMs shown in Figure 2 and Figure 4 are all 13B in parameter count, while there is no published parameter count for GPT-3.5 or Bard. Therefore it is possible that the difference seen here is attributable to differences in model size rather than details in the model's development (e.g., OpenAI and Google's undisclosed training recipes).
W4. The English of this manuscript could use some improvement. E.g., some phrases were not idiomatic, which hinders comprehension (e.g., "major-level theorems"). I suggest the authors request editing help from a native English-speaking colleague.

**Questions:**

Q1. From each of the 120 datasets, how did the authors select the particular examples to be included in FLASK? This question is important because it affects whether the instances in FLASK are representative or tend to be outliers in certain aspects.
Q2. Do the authors plan to make the FLASK dataset open source? This is not addressed in the manuscript.

---

> ### Author Response · Authors · 2023-11-15
> **Rebuttal by authors**
>
> Thank you for taking the time to review our paper. Our responses are shown below.
>
> ---
> **W1. Arbitrariness in creating the list of 12 fine-grained skills**
>
> We acknowledge that some arbitrariness exists in defining the 12 fine-grained skills. To establish the 4 abilities and 12 fine-grained skills, we drew upon Rogers et al's (2021) categorization and modified it—for example, changing 'Retrieval' to 'Background Knowledge'—to suit the evaluation of language model alignment. We suggest that future work could refine this categorization, possibly by adding or merging specific skills based on distinct use cases. In this case, FLASK could serve as a foundational framework for fine-grained evaluation. Regarding the specific concern about the arbitrary nature of some skills, we provide the following responses:
>
> 1. *Could "readability" be singled out as an "essential skill"?*
>
> We agree that skills such as Readability or Comprehension could be apply all instances. However, our goal of the fine-grained evaluation based on skill sets is to focus on skills that are *truly* essential for the instance. For example, logical problems might not need the skill of readability as much as writing tasks such as “Write a template for a First-Person LinkedIn profile summary.”. To achieve a balance, we found that annotating the top three (K=3) relevant skills per instance was optimal. This approach prevented us from labeling every instance with universal skills like readability while ensuring that instances truly needing those skills were accurately identified.
>
>
> 2. *Could metacognition be labeled in a model-independent manner?*
>
> We acknowledge that the concept of metacognition could be dependent on the specific model being evaluated. However, since language models are evaluated by text-based LLMs or humans in this work, our reference standards are aligned with the capabilities of GPT-4.  Therefore, instances that require other modalities (ex) Do you like the taste of pizza?) or predicting future events (ex) If bitcoin has gone up in value over the last twenty years, what do we know will happen in the next twenty years? Will a 100 trillion parameter deep learning model be trained before 2026?) are mainly annotated to include metacognition.
>
>
> We have added the details regarding the specific skills in the revised paper. (Appendix E)
>
>
> Reference:
> Rogers et al (2021), QA Dataset Explosion: A Taxonomy of NLP Resources for Question Answering and Reading Comprehension
>
>
>
> ---
> **W2. Missing domains and multiple domains**
>
> 1. Missing domains
>
> We divided the domains into 10 domains and 38 sub-domains as shown in Table 6 in the Appendix. Domains such as laws and finance are classified as Social Science domains according to the table and the travel domain should be classified as Culture considering the similarity with the sub-domains of the Culture domain. During domain annotation, we also provide the sub-domain information in the prompt to specify the boundary between various domains.
>
>
> 2. Multiple domains
>
> We allow multiple domain (up to two) annotations during the domain annotation process by specifying in the annotation instruction that two domains may be labeled when it is hard to annotate only a single domain.
>
> ---
> **W3. Comparison of open and closed source LLMs**
>
> We acknowledge that it is hard to draw conclusions for closed-source LLMs since the model details are not provided. We also agree with your point that the performance gap between open-sourced and closed-source LLMs might be due to the differences in model size rather than specific training techniques. We have revised the paper by deleting hypothetical interpretations regarding the comparison between open-sourced and closed-source LLMs to avoid misinterpretations.
>
> ---
> **W4. English presentation**
>
> We have updated the paper to enhance the English expressions. Thank you for your suggestion.
>
> ---
> **Q1. Instance selection criteria for each dataset consisting of FLASK**
>
> For the construction of the FLASK evaluation set, we prioritized diversity to prevent bias towards any particular distribution. In single-task datasets, we limited the number of instances to a maximum of 20. For multi-task datasets such as Self-Instruct and WizardLM, we observed the inherent diversity between instances and did not impose this restriction. For multi-task datasets such as MMLU and BBH, we sampled a single instance per sub-dataset since there was a significant overlap of required skills, domain, and difficulty between the same sub-dataset. The full distribution of each dataset consisting of FLASK is presented in Table 12 in the Appendix.
>
> ---
> **Q2. Plan for open-sourcing the FLASK dataset**
>
> Yes, we are planning to open-source the dataset, the metadata annotation, the evaluation pipeline (code implementation), and the interactive visualization demo page for analysis.

---

### Official Review · Reviewer_BpKX · 2023-11-01

**Soundness:** 3 good
**Presentation:** 3 good
**Contribution:** 3 good
**Rating:** 6
**Confidence:** 4

**Summary:**

This study introduces a fine-grained evaluation protocol called FLASK, aimed at assessing the alignment capabilities of language models (LMs). FLASK emphasizes four primary aspects: Logical Thinking, Background Knowledge, Problem Solving, and User Alignment, that are further divided into 12 fine-grained skills to facilitate comprehensive LM evaluation. The authors design specific scoring rubrics for each task, providing guidance to human evaluators or model-based evaluators in accurately measuring the skill level of LMs.

Moreover, this research proposes an approach that utilizes GPT-4 as an evaluation model to automate the evaluation process. The authors conduct extensive experiments to demonstrate the effectiveness of their proposed method, evaluating its robustness, reliability, and the correlation between human-based and model-based evaluations, etc.

**Strengths:**

1. This paper introduces FLASK, a new evaluation framework designed to assess the performance of language models in a fine-grained manner. FLASK encompasses a diverse range of essential skills for language models, such as logical correctness, commonsense understanding, comprehension, harmlessness, and more.

2. By employing task-specific scoring rubrics, both human evaluators and evaluation models can accurately assess the performance of language models and provide qualitative analyses based on the rules outlined in the rubric descriptions.

3. The authors conducted comprehensive experiments to showcase the superiority and reliability of their proposed method. These experiments provide strong evidence supporting the effectiveness and robustness of the proposed approach.

**Weaknesses:**

1. The evaluation of large language models using only 1740 instances is inadequate to ensure thorough verification. Furthermore, the limited number of instances makes it challenging to encompass a wide range of domains, thereby diminishing the effectiveness of the proposed method.

2. The experimental results presented in Table 4 reveal a relatively low level of agreement among different evaluation models. It is possible that a large language model performs well in a specific skill but receives a lower score due to the evaluation model's inability to effectively address problems within the current domain or skill set.

**Questions:**

Does the FLASK-HARD set encompass all 12 skills utilized in this study? Furthermore, what is the distribution of these 12 skills within the FLASK-HARD set?

---

> ### Author Response · Authors · 2023-11-15
> **Rebuttal by authors**
>
> Thank you for taking the time to review our paper. Our responses are shown below.
>
> ---
> **W1: Inadequate number of evaluation instances**
>
> We acknowledge that the number of evaluation instances might not be sufficient. We controlled the FLASK whole set to be around 1800 instances to enable academic budget evaluation using model-based evaluation (GPT-4) or human-based evaluation. However, one of the advantages of the FLASK metadata annotation process is that it could be automated by utilizing LLMs as described in Section 3.2. Therefore, the evaluation instances could be dynamically expanded without extensive labeling processes. This indicates that although we limited the number of instances within an academic budget, the dataset could be easily expanded for increased budget (e.g. industry budget).
>
> ---
> **W2: Low level of agreement among evaluation models**
>
> The low level of agreement among evaluation models (GPT-4, Claude, GPT-3.5) in Table 4 is likely due to the discrepancy in the evaluation capabilities between GPT-4 and the other two models. Table 1 in the paper shows that GPT-4's correlation with human evaluators is significantly higher compared to Claude or GPT-3.5.  Despite acknowledging the limitations inherent in the capabilities of the evaluation models, we suggest that employing better underlying models for evaluation like GPT-4, or language models specifically fine-tuned for reliable evaluation (Kim et al, 2023), could mitigate this issue.
>
>
> Reference:
> Kim et al (2023), Prometheus: Inducing Fine-grained Evaluation Capability in Language Models
>
> ---
> **Q1: Distribution of the skills within the FLASK-Hard set**
>
> Yes, FLASK-HARD encompasses all 12 skills. We have added the distribution of 12 skills within the FLASK-HARD set below and also in the revised paper (Figure 25). We can observe that the skill distribution of FLASK-Hard is similar to the distribution of the whole FLASK evaluation set.
>
>
> Logical robustness: 4.9%
>
> Logical correctness: 10.1%
>
> Logical efficiency: 6.0%
>
> Factuality: 10.1%
>
> Commonsense Understanding: 15.0%
>
> Comprehension: 22.1%
>
> Insightfulness: 4.5%
>
> Completeness: 6.7%
>
> Metacognition: 7.1%
>
> Readability: 4.5%
>
> Conciseness: 4.5%
>
> Harmlessness: 4.5%

---

> > ### Comment · Reviewer_BpKX · 2023-11-18
> >
> > I appreciate the responses given by the authors. addressed my concerns

---

### Meta-Review · Area_Chair_Xmn4 · 2023-12-05

**Metareview:**

The reviewers seem to appreciate the fine-grained nature of the proposed evaluation framework and agree that it will be useful to the community. They also comment on comprehensive experiments conducted by the authors. However, they also raise some concerns primarily around evaluation and results given that the test set comprises only 1,700 instances that may be inadequate for today's standards (and multi-faceted evaluations needed), the categorization of the skills, and the fact that all annotators are senior undergraduate students.

The authors respond to the reviewers adequatly but I don't find all their responses convincing, therefore I'm recommending accepting as a poster or spotlight paper.

**Justification For Why Not Higher Score:**

I didn't find all the author responses convincing, there are some points for improvement in this work.

**Justification For Why Not Lower Score:**

This paper has potential of good impact, and is very timely.

---

### Decision · Program_Chairs · 2024-01-16

Accept (spotlight)